# The Role of Rab Proteins in Mitophagy: Insights into Neurodegenerative Diseases

**DOI:** 10.3390/ijms24076268

**Published:** 2023-03-27

**Authors:** Adeena Shafique, Martina Brughera, Marta Lualdi, Tiziana Alberio

**Affiliations:** Department of Science and High Technology, Center of Research in Neuroscience, University of Insubria, I-21052 Busto Arsizio, VA, Italy

**Keywords:** vesicular trafficking, neurodegeneration, Ras analog in brain (Rab), mitophagy

## Abstract

Mitochondrial dysfunction and vesicular trafficking alterations have been implicated in the pathogenesis of several neurodegenerative diseases. It has become clear that pathogenetic pathways leading to neurodegeneration are often interconnected. Indeed, growing evidence suggests a concerted contribution of impaired mitophagy and vesicles formation in the dysregulation of neuronal homeostasis, contributing to neuronal cell death. Among the molecular factors involved in the trafficking of vesicles, Ras analog in brain (Rab) proteins seem to play a central role in mitochondrial quality checking and disposal through both canonical PINK1/Parkin-mediated mitophagy and novel alternative pathways. In turn, the lack of proper elimination of dysfunctional mitochondria has emerged as a possible causative/early event in some neurodegenerative diseases. Here, we provide an overview of major findings in recent years highlighting the role of Rab proteins in dysfunctional mitochondrial dynamics and mitophagy, which are characteristic of neurodegenerative diseases. A further effort should be made in the coming years to clarify the sequential order of events and the molecular factors involved in the different processes. A clear cause–effect view of the pathogenetic pathways may help in understanding the molecular basis of neurodegeneration.

## 1. Introduction

Vesicular trafficking alterations have been implicated in the pathogenesis of several neurodegenerative diseases (NDs) as likely early events [1,2]. Two main branches of vesicle trafficking exist in neuronal cells: the canonical shuttling of cellular components and the mobilization of synaptic vesicles. Both processes are tightly regulated and orchestrated by multiprotein complexes. Their malfunction causes profound alterations in cellular homeostasis.

As for canonical vesicle trafficking, the three central pathways are: (i) the secretory pathway, (ii) the endocytic pathway, and (iii) specialized/organelle-specific vesicular transports [3]. The secretory pathway, through which proteins and lipids move from the ER through the Golgi to the extracellular environment, involves two main processes, namely, the anterograde (from ER to Golgi) and the retrograde (from Golgi to ER) transports, which are mediated by COPII and COPI coatomers, respectively. The endocytic pathway, in which several cargos can be internalized by cells, involves the formation of early and late endosomes, which eventually fuse with lysosomes; in this process, several proteins assist vesicle formation, e.g., clathrin, caveolins, adaptors, GTPase dynamins, Ras analog in brain (Rab) proteins, and vacuolar protein sorting (VPS) proteins. Eventually, several specialized vesicle transports have been recently discovered, which mirror the high dynamism and social behavior of cellular organelles [4]. The endoplasmic reticulum (ER), Golgi apparatus, lysosomes, peroxisomes, and mitochondria are known to physically interact and exchange components thanks to both contact sites and vesicles formation; this allows organelles to modulate specific cellular functions, pathways, and metabolism. As an example, mitochondria are known to communicate with the ER, Golgi apparatus, lysosomes, lipid droplets, peroxisomes, and melanosomes [5]. The contact sites between mitochondria and the ER (i.e., mitochondria-associated membranes, MAMs) are important hubs for lipid trafficking, Ca^2+^ homeostasis, ER stress, apoptosis, and macroautophagy. On the other hand, the mitochondria-lysosome axis leads to the formation of vesicles of mitochondrial origin (i.e., mitochondria-derived vesicles, MDVs), whose generation and release are central to the disposal of defective components without triggering the mitophagic process. However, the content of MDVs is still mostly unknown; some components seem to play a role in the activation of inflammatory processes and are being investigated as candidate plasma biomarkers for NDs [6,7].

As for synaptic vesicles, neuronal synapses are specialized subcellular compartments, where the cycles of exocytosis and endocytosis are regulated by a complex molecular machinery, including several proteins, e.g., SNAREs, synaptotagmin, synaptophysin, and some small GTPases belonging to the Rab family (mainly Rab3, Rab5, and Rab27) [8]. Synaptic vesicles are classified in pools based on their function: they can either be part of the reserve pool or be involved in active exo- and endocytotic cycles. The mobilization of vesicles from one pool to another is an ATP-dependent process, fueled by the mitochondria that are resident in the synapses. Thus, synapses which lack mitochondria display marked defects in vesicular trafficking that are mirrored by deficits in neuronal functionality [9]. This further supports a clear link between mitochondrial dysfunction, which is a crucial pathogenetic process in many NDs, and defects in vesicular trafficking.

Mitochondria are central organelles in all cell types, being responsible for energy production in the form of ATP, the biosynthesis of amino acids and steroids, β-oxidation of fatty acids, the maintenance of cytosolic calcium homeostasis, the production and modulation of reactive oxygen species (ROS), and the triggering of the apoptotic cell death program. Moreover, they are dynamic organelles that are transported on cytoskeletal proteins and organized in networks, which are continuously remodeled by fusion and fission events [10]. This allows for both the exchange of mitochondrial components and the elimination of damaged parts to maintain a functional pool of healthy mitochondria. Damaged mitochondria must be disposed of through a macroautophagic pathway called mitophagy. Canonical mitophagy involves PTEN-induced kinase 1 (PINK1) and Parkin proteins, which are responsible for the recruitment of the molecular machinery that guides the formation of the autophagosome and the subsequent fusion with lysosomes [11]. When mitophagy is hampered, dysfunctional mitochondria accumulate within cells, thus increasing ROS concentration and oxidative load in general. Considering the huge energy demand and the terminally differentiated state of neuronal cells, it appears evident that any alterations to physiological mitochondrial homeostasis can have devastating effects on neuronal survival.

Nowadays, there is convincing evidence of impaired mitochondrial function as a cause rather than a consequence of the neurodegenerative process in several NDs, such as Alzheimer’s disease (AD), Huntington’s disease (HD), amyotrophic lateral sclerosis (ALS), and Parkinson’s disease (PD). For instance, in the frame of PD etiopathology the vast majority of genetic PD-related loci are associated with mitochondria, and aberrant mitochondrial function has emerged as central also in sporadic PD [12]. Recently, vesicular trafficking defects have also been highlighted as important pathogenetic factors [13]. Alpha-synuclein, the major component of toxic protein aggregates in PD, has been shown to directly bind the SNARE protein VAMP2 and to be critical for the proper assembly of the SNARE complex [14]. LRRK2 protein, whose gene mutation is linked to autosomal dominant inherited familial PD, has been demonstrated to co-localize with Rab5 on endocytic vesicles [15] and to be the kinase of several Rab proteins [16,17]. The substrate repertoire of Parkin, which acts as an E3 ubiquitin ligase and is found to be mutated in autosomal recessive juvenile PD, includes several proteins involved in vesicular trafficking, such as Synphilin1 and Synaptotagmin XI [18]. In addition, recent genome-wide association studies (GWASs) have proposed new loci associated with an increased risk of PD, which encode proteins involved in vesicular trafficking, such as VPS35, Synaptojanin1, VPS13C, Rab39B, Rab7L1/Rab29, and Synaptotagmin XI [13,19,20].

Here, we provide an overview of major findings in recent years highlighting the role of a family of proteins involved in vesicle trafficking, namely, Rab proteins, in altered mitochondrial dynamics and mitophagy, which are characteristic of NDs (Figure 1).

## 2. Mitochondrial Dynamics Altered in Neurodegeneration

Mitochondria are double-membraned organelles strictly involved in the regulation of cellular homeostasis and metabolism. Among the most common known functions, such as ATP production, ion homeostasis, biosynthesis of precursors for macromolecules, and management of toxic metabolic by-products, mitochondria are emerging as being more and more relevant to regulating signaling pathways and cellular stress responses [21]. They are interconnected entities that constitute branched tubular networks, which are not fixed structures; rather, they are highly dynamic and their shape is deeply affected by perturbations to cellular functions [22].

Mitochondrial dynamics collectively refer to all the processes involved in the morphological changes of the mitochondrial network, and they include fusion, fission, mitophagy, and mitochondrial transport [23]. These processes are deeply interrelated and part of the quality control of mitochondrial organelles. The fine tuning of fusion and fission dynamic transitions is orchestrated mainly by the dynamin-related family of large GTPases, and the resulting balance between the two processes is responsible for the number and size of mitochondria [10]. Mitophagy is the process that allows dysfunctional mitochondria to be degraded. It can be triggered by different stimuli and completed by distinct pathways [24]. The altered balance of the abovementioned processes often results in the intracellular accumulation of dysfunctional organelles, thus contributing to the onset and progression of several NDs.

### 2.1. Fusion and Fission

The fusion process occurs when two mitochondria merge both their outer and inner membranes. These two distinct steps are regulated by two different fusion machineries: the membrane-anchored dynamin family members mitofusin 1 (Mfn1) and mitofusin 2 (Mfn2) allow outer mitochondrial membrane (OMM) fusion, while the Dynamin-like 120 kDa protein (OPA1) controls inner mitochondrial membrane (IMM) fusion [10,25]. Fission events, on the contrary, produce two daughter mitochondria and are mediated by a cytosolic dynamin family member called dynamin-related protein 1 (DRP1), which is recruited from the cytosol to mitochondria, oligomerizes to form ring structures around mitochondrial membranes, and induces membrane constriction after GTP hydrolysis [10,25]. Impairment of these two processes can be the triggering cause or a consequence of the neurodegenerative process.

Autosomal dominant optic atrophy (DOA) served as a model to understand the function of OPA1 and investigate the consequences of fusion defects. By exploiting control and patient-derived fibroblasts carrying biallelic mutations in *OPA1*, Liao and colleagues [26] observed increased fragmentation and perinuclear clustering of the mitochondrial network, together with lower mtDNA levels and increased mitophagy induction. Most frequently, mitochondrial fusion defects in NDs are due to lower abundance or impaired activity of OPA1 rather than genetic mutations. In fact, in a PD mouse model recapitulating complex I defects, treatment with the mitochondrial toxin 1-methyl-4-phenyl-1,2,3,6-tetrahydropyridine (MPTP) induced an overall reduction in OPA1 protein levels [27]. Further in vitro experiments exploiting MPP^+^-induced toxicity in SH-SY5Y neuroblastoma cells confirmed the relevant role of OPA1 in mitochondrial morphology, since, after treatment, the mitochondria appeared swollen with disrupted cristae. A higher level of ROS is normally generated by complex I deficiency; therefore, the authors explained the mitochondrial remodeling observed with the oxidative-dependent disruption of OPA1 oligomeric complexes, known to be involved in maintaining cristae integrity [27]. Another disease, namely, ataxia with oculomotor apraxia type 1 (AOA1), further helped in the elucidation of OPA1′s role in mitochondrial dynamics. This disease is caused by mutation in aprataxin (*APTX*), a DNA ligase “proofreader”. In the APTX KO model, lower levels of OPA1 have been related to a fragmented mitochondrial network and deficient mitophagy [28].

Mutations in *MFN1* and *MFN2* genes have been linked to impaired fusion, too. Charcot-Marie-Tooth type 2 A pathology, a hereditary peripheral axonal neuropathy, is caused by mutations in *MFN2* and it is characterized by the presence of a fragmented mitochondrial network in patient-derived fibroblasts [29]. In fact, in a recent work by Zhou and colleagues [30], mitochondria from mouse embryonic fibroblasts (MEFs) expressing the MFN2^R94Q^ mutant, besides being depolarized, were characterized by a decreased length/width ratio, as a measure of mitochondrial fusion.

In AD models, alteration of the mitochondrial network has been frequently reported [31]. Harland and colleagues [32] exploited transgenic mice overexpressing Mfn2 (TMFN) and injected them with lipopolysaccharide (LPS) to trigger neuroinflammation, typically associated with AD. Mfn2 overexpression in TMFN mice was correlated with reduced mitochondrial fragmentation, and mice displayed less LPS-induced inflammation and “behavioral sickness”. Fission inhibition was demonstrated to have a beneficial effect, also, in another AD model obtained by the overexpression of amyloid-β oligomers. Indeed, the mitochondrial fission inhibitor *mdivi-1* alleviated both synapse loss in cultured hippocampal neurons and cognitive impairment in mice [33].

Rearrangements of the mitochondrial network are needed to deal with mitochondrial transport, bioenergetic needs, mitosis, and maintenance of genetic homogeneity within the mitochondrial population [23]. In addition to this, mitochondrial dynamics are also involved in adaptation to cellular dyshomeostasis, and enhancement of either fusion or fission events are exploited as mechanisms to cope with stress. According to the extent of stress-induced damages, mitochondria can restore their normal function after the selective excision of defective parts via MDVs. However, if stress stimuli are prolonged in time and mitochondria are beyond repair, mitophagy occurs [34].

As reported above in several NDs, alterations affecting fusion and fission processes ultimately lead to the disruption of the fine-tuned equilibrium between the two. This highlights how the well-being of neurons is strictly linked to mitochondrial dynamics. This evidence hints at future steps directed at further studying and modulating the key players of fusion and fission as therapeutic targets.

### 2.2. Mitophagy

There are two main mechanisms through which dysfunctional mitochondria can be labeled and removed: ubiquitin-dependent mitophagy and receptor-mediated mitophagy [35].

In ubiquitin-dependent mitophagy, one of the most studied pathways involves PINK1 and Parkin-mediated ubiquitination [36]. Under physiological conditions, PINK1 is targeted to mitochondria, where it is cleaved by the intramembrane protease presenilin associated rhomboid like (PARL); then, proteolytic fragments are degraded by the cytosolic ubiquitin proteasome system [37,38]. When mitochondrial depolarization occurs, PINK1 is not transported inside mitochondria and aggregates on the OMM, where it phosphorylates Parkin. Once Parkin (an E3 ubiquitin ligase) is activated, it ubiquitinates outer membrane protein targets, such as Voltage-Dependent Anion Channels (VDACs), Mfn1, Mfn2, and Miro, promoting mitochondria engulfment in autophagosomes and subsequent lysosomal degradation [11,24].

Mutations in the *PARK2* gene have been known, among others, to be involved in the pathogenesis of PD since the late 1990s [39], but only several years after this discovery was their pivotal role in mitophagy described [36]. In *PARK2*-mutated fibroblasts, derived from early-onset PD patients, mitochondria are depolarized but do not display accumulation of PINK1, thus impeding the proper triggering of mitophagy [40]. Later, PINK1, whose gene is mutated in a familial form of PD, was also associated with mitophagy regulation [41,42,43]. Recently, two new mutations were discovered in *PINK1* which increase the susceptibility to develop sporadic PD. These mutations cause different substitutions at the same site: PINK1^G411S^ displays a higher level of autophosphorylation, which hampers its kinase activity toward ubiquitin (Ub), while PINK1^G411A^ shows an enhanced kinase activity [44]. Parkin-mediated attachment of Ub chains, in fact, relies on a positive feedback cycle of PINK1-mediated phosphorylation of ubiquitin (p-Ser65-Ub) of OMM substrates. This molecular tag recruits Parkin from the cytosol, so that ubiquitin chains keep forming on damaged mitochondria [37]. In a recent, high-content phenotypic screening, p-Ser65-Ub was used as a specific biomarker of PINK1/Parkin-dependent mitophagy to identify compounds able to boost Parkin-mediated ubiquitination, and therefore mitophagy, in PD-patient-derived fibroblasts [45].

In sporadic and familial forms of PD, impaired mitophagy is a common feature [46]. In the SH-SY5Y neuroblastoma cellular model of altered dopamine homeostasis, which recapitulates the early stages of PD pathology, the mitochondrial network appeared hyperfused and PINK1 was not recruited at depolarized mitochondria, thus impairing their disposal [47]. Lower levels of VDAC1 and VDAC2 have also been described in the same model, which may justify a reduced tagging of dysfunctional mitochondria [48]. Studies on T415N mutant Parkin (associated with PD) highlighted how the inability of Parkin to monoubiquitinate VDAC1 impairs mitophagy. Indeed, after mitophagy induction with the ionophore carbonyl cyanide 3-chlorophenylhydrazone (CCCP), dysfunctional mitochondria were not eliminated due to the absence of monoubiquitinated VDAC1. Of note, this mutation in Parkin abolishes VDAC1 monoubiquitination but still retains its polyubiquitination activity, thus promoting apoptosis by increasing mitochondrial calcium uptake [49]. In this view, VDACs have also been recently considered valuable targets for PD treatment [50]. Mutations in either Parkin or OMM proteins’ genes can be responsible for impaired ubiquitination of OMM target proteins and subsequent deficient mitophagy. Indeed, in both *PARK2*-mutated fibroblasts and Mfn2 KO MEFs, Mfn2 ubiquitination was not observed [51]. Moreover, mitochondria-ER contact sites (MERCs), which are important for the recruitment of membranes enveloping damaged mitochondria, were absent. This was further sustained by the reverted locomotor deficit phenotype associated with an in vivo *Drosophila* model of PD, after the expression of an ER-mitochondria synthetic linker [51].

In the context of AD, the presence of Aβ and tau proteins has been linked to ultrastructural abnormalities of mitochondria accompanied by aberrant modulation of mitophagy [52]. Cummins and colleagues [53] showed impaired recruitment of Parkin to mitochondria after CCCP treatment in N2a murine neuroblastoma cells expressing human P301L mutant tau. Similarly, the overexpression of the amyloid-beta protein precursor (AβPP) caused an increase in Parkin levels, but the protein was incorrectly localized, and the recruitment of PINK1 was impaired [54]. Moreover, AβPP overexpression was able to increase the levels of the PINK1 processed form (∆1-PINK1), recently described as the form that sequesters cytosolic Parkin [55]. In line with this, investigation of serum proteins from patients suffering from AD, mild cognitive impairment (MCI), and “mixed” dementia (MD) showed lower levels of Parkin [56]. Moreover, Goiran and colleagues [57] recently demonstrated that the amyloid-β protein precursor intracellular domain (AICD) was able to act as a positive modulator of PINK1 transcription. Collectively, these results support the hypothesis that *PINK1* expression is activated in AD, as a proper attempt to trigger mitophagy, but the process fails due to Parkin sequestration in the cytoplasm and/or inefficient recruitment to mitochondria. According to this possible explanation, overexpression of Parkin in Aβ-treated HEK293 cells attenuated mitochondrial dysfunction [58].

After PINK1/Parkin recruitment and activation, the following stage in the mitophagic process is the amplification of the ubiquitination reaction. This step is needed to recruit cargo adaptors and activate the autophagic machinery, which oversees the engulfment of damaged mitochondria within autophagosomes and subsequent degradation by lysosomes [59,60]. Among adaptors, the scaffold proteins OPTN, NDP52, p62, CALCOCO2, NBR1, and TAX1BP1 have been associated with Ub-mediated autophagy, where they act as a bridge between the ubiquitinated cargoes and autophagosomal membranes [38]. Some proteins whose functions are altered in NDs are known to play a role in this step of the mitophagic process. As an example, huntingtin protein (HTT) is known to interact with OPTN and CALCOCO2 receptors. In HD, the presence of the mutated polyglutamine (polyQ) HTT affects these interactions, thus impairing the subsequent recruitment of LC3-bound membranes to dysfunctional mitochondria [61].

As mentioned above, another way also exists to recognize mitochondria that must be disposed of, independently of PINK1/Parkin recruitment. In receptor-mediated mitophagy, OMM proteins act directly as autophagy receptors, linking mitochondria to autophagosomal membranes. BNIP3, BNIP3L/NIX, and FUNDC1 are among the most common autophagy receptors, and they are primarily involved in hypoxia-induced mitophagy [62]. Upon stress stimuli, BNIP3L/NIX receptors are inserted into the OMM where they shift to the active form after dimerization [63]. BNIP3 is a pro-apoptotic member of the Bcl-2 family, whose expression levels are influenced by stress stimuli [64]. It was demonstrated that BNIP3 directly interacts with OPA1 complexes [65]. In OPA1 knockdown (KD) cellular models, reflecting DOA pathology, the reduced presence of OPA1 led to a lower level of BNIP3 that resulted in a lesser extent of mitophagy. Mitophagy induction was rescued by boosting the expression of BNIP3 upon deferoxamine treatment, a mimetic of hypoxia [66].

As a later step, after the recognition of damaged mitochondria, the autophagy cascade can be summarized in sequential steps: (i) nucleation of the initial membrane; (ii) regulated expansion of the isolation membranes, which are now referred to as phagophores; (iii) formation of a closed, double-membraned vesicle called an autophagosome; and (iv) subsequent vesicle fusion with the lysosomes. Autophagosome formation is regulated by autophagy-related (ATG) proteins. PtdIns(3)P-rich (PI3P) membrane domains on the ER are the preferential autophagosome formation sites [67] due to their ability to change shape and elongate [68]. Phagophore formation is started by the ULK1 complex [67,69], which phosphorylates the VPS34 kinase [70], then elongation and expansion occur [71]. These steps can be impaired in NDs. For instance, the activation of ULK1 can be dysfunctional in HD. Indeed, HTT plays a crucial role in guiding the formation of the ULK1 complex, since it acts as a scaffold by facilitating the interaction between cargo receptors and the initiator complex. HTT mutations affect the binding with ULK1, which in turn is less prone to detachment from the MTOR complex 1 (MTORC1) that negatively regulates the early steps of autophagy [61].

Sequential steps of phosphorylation of several ATG proteins are later responsible for the expansion and maturation of the autophagosome [72]. For instance, the activation of the ATG12-ATG5-ATG16L complex has a key role in LC3 lipidation, which is crucial for autophagosome maturation, cargo capture, engulfment of the organelle, and lysosomal targeting [68,69,73]. Additionally, these later steps of autophagosome maturation were found to be aberrantly regulated in NDs. Indeed, patients affected by AD, MD, and MCI showed significantly reduced circulating levels of ATG5, and this was proposed, together with Parkin, as a circulant biomarker to monitor patients with cognitive decline [56]. Related to AD, Martìn-Maestro and colleagues recently reported that skin fibroblasts from sporadic AD patients showed alterations in lysosomes and autophagy upon CCCP treatment, accompanied by an increase in oxidized and ubiquitinated proteins. In particular, fibroblasts displayed a small number of lysosomes with altered pH (increased). A clear mitophagy impairment was also present, due to diminished Parkin and insufficient vesicle induction, which was recovered by Parkin overexpression [54]. In another work, Wei and colleagues [74] investigated the relationship between estrogen receptor β (Erβ), autophagy, and Aβ degradation in AD. Indeed, they observed enhanced Aβ degradation via modulation of Erβ, which was able to boost lysosomal activity (higher LC3 levels detected) and ATG7 recruitment. Thus, the overexpression of ERβ exerts a neuroprotective effect through the enhancement of autophagy–lysosomal activity.

To sum up, alterations in all stages of mitochondrial quality-control processes and mitochondrial disposal via mitophagy can be implicated in the onset and progression of NDs, as summarized in Figure 2. Disentangling the complexity of all the involved molecular factors, their interactions, and participating pathways will shed light on the pathobiology of NDs and suggest novel candidates for therapeutic interventions.

One family of proteins that looks promising for the identification of novel therapeutic targets is the Rab protein family. Indeed, a strong involvement of Rab proteins in orchestrating the vesicle trafficking related to mitophagy has been recently unveiled, with important implications for NDs.

## 3. Rab Proteins at the Crossroads of Mitochondrial Dysfunction and Neurodegeneration

Rab proteins represent the largest family of Ras-like GTPases. They are largely distributed in eukaryotic cells with more than 70 members identified in mammals and 11 homologs in budding yeast [75,76]. Rab proteins exist within cells as molecular switches and shift between an active GTP-bound form, which associates with intracellular membranes, and an inactive GDP-bound conformation, which is present in the cytosol. These small GTPases require prenylation at their C-terminal cysteine residues to associate with membranes. This process is aided by geranylgeranyl transferases (GGTs) and Rab escort proteins (REPs). When membrane-bound, Rab proteins are activated by the exchange of GDP with GTP, and this is catalyzed by guanine nucleotide exchange factors (GEFs). In their active state, Rab proteins perform a myriad of functions, including vesicular budding, trafficking, and fusion with target membranes. On delivering cargo, Rabs undergo hydrolysis of the bound GTP to GDP with the help of GTPase-activating proteins (GAPs). In the inactive state, Rab proteins are retrieved and sequestered in the cytosol by GDP dissociation-inhibitor (GDI) proteins [77,78].

Rab proteins have been linked to multiple diseases, including NDs, cancer, and diabetes [79]. Among NDs, Rab proteins have been linked to dementia, AD, PD, HD, and ALS. In humans, 24 Rab proteins were found to be involved in the central nervous system [80], where many of them perform specialized neuronal functions [81]. More recently, the involvement of altered vesicle trafficking, particularly that which is Rab protein-mediated, has been highlighted in ND pathobiology. This finding has also proven to be a challenge, as studies are endeavoring to determine whether Rab proteins are causes or effects in the pathogenesis of NDs.

As Rab proteins are master regulators of vesicle trafficking, it is not surprising that they are emerging as key players in the molecular pathways of PD through a complex interplay with several PD-related genes. The question to be addressed is which Rab proteins are relevant to the mechanisms involved in PD, and to what extent. The most striking evidence of Rab proteins’ involvement came from the discovery of genetic alterations in familial forms of PD. Rab39B has a high brain-specific and exclusively neuron-specific expression [82]. Mutations in *Rab39B* resulted in a complete loss of protein expression and function and X-linked intellectual disability with early-onset PD [83]. Moreover, in Rab39B knockout mice, deficiency in Rab39B led to impaired learning and memory and impaired basal autophagic flux [84]. Another Rab gene, *Rab29* (*Rab7L1*), has emerged as a leading candidate for sporadic PD, being one of the five genes that belongs to the *PARK16* locus. Its role in PD pathogenesis is not completely understood; however, several studies have pointed to its upstream regulatory role in relation to LRRK2, whereby Rab29 increases LRRK2 activity and its localization to the trans-Golgi network [16,85,86]. *LRRK2* mutations are the most common causes of familial PD and are also implicated in sporadic PD. The link between LRRK2 and Rab proteins came from the discovery that LRRK2 phosphorylates a wide subset of them, including Rab1A, Rab1B, Rab3, Rab5, Rab8A, Rab10, Rab12, Rab29, Rab35, and Rab43 [86]. All pathological *LRRK2* mutations lead to hyperactivation of the protein, followed by abnormal Rab phosphorylation, which is abrogated upon pharmacological inhibition of LRRK2. Indeed, clinical trials are currently ongoing using small molecules as inhibitors of LRRK2 kinase activity, with potential beneficial effects in both familial and sporadic PD cases. As a downstream effect of LRRK2 inhibition, the restoration of physiological activity of Rab proteins is crucial to explanations of the therapeutic effects of these drugs [87]. An integrated omics analysis revealed dysregulation of the endocytic pathway in iPSC-derived dopaminergic neurons carrying the G2019S mutation in LRRK2. This dysregulation could be due to changes in the activity of several Rab proteins, including Rab5B, Rab7, and Rab10 [88].

Several studies have also established the role of Rab proteins in modulating α-synuclein biology in different model systems. Rab5, an early endosomal protein, is integral to the biogenesis of early endosomes [89]. It has been reported in the endocytosis of α-synuclein leading to neuronal death [90]. Another endosomal Rab, Rab7, was shown to clear α-synuclein aggregates. In HEK293 cells and *Drosophila melanogaster*, overexpression of Rab7 led to the clearance of α-synuclein aggregates and reduced cell death, thereby showing its protective role. This protective effect was specific to Rab7, as Rab5, Rab9, and Rab23 failed to rescue the phenotype [91]. Moreover, over-expressed α-synuclein was found to interact with dynein and induced an increase in the levels of GTP-bound Rab5 and Rab7 [92]. Eventually, overexpression of Rab11, physiologically involved in endosomal recycling, has also been shown to regulate α-synuclein dynamics in *Drosophila* models of α-synuclein toxicity [93].

As mitochondrial dysfunction makes a clear contribution to NDs, particularly PD, it is imperative to understand the pathways that regulate mitochondrial health. It is quite evident that mitochondrial quality-control pathways involve an intricate network of proteins and pathways, many of which have not been clearly understood and identified yet. For quite some time, macroautophagy was believed to be the only mechanism for degrading organelles, including mitochondria. Many studies are building up evidence that brings Rab proteins to the forefront in the clearing of damaged mitochondria via different pathways. Studies have also reported that alterations to Rab proteins related to their phosphorylation states, expression levels, and localization can lead to defects in mitophagy. Herein, we have presented an overview of the canonical and non-canonical mitophagy pathways in which specific Rab proteins play key roles.

### 3.1. The Role of Rab Proteins in Canonical PINK1/Parkin-Mediated Mitophagy

Rab GTPases are required at several steps of the classical mitophagy pathway. Specifically, endolysosomal Rab5 and Rab7 have been highlighted as key players in this process. The first evidence signifying the role and involvement of Rab proteins in mitophagy came from Yamano and colleagues [94]. Their study demonstrated the involvement of Rab7 in the mitochondrial quality-control pathway. Indeed, Rab7 GAPs TBC1 domain family member 15 (TBC1D15) and TBC1 domain family member 17 (TBC1D17) regulate Rab7 activity and mediate mitochondrial sequestration in autophagosomes. This study particularly highlighted the role of Rab7 in mito-autophagosome biogenesis. In their follow-up study [95], Yamano and colleagues further demonstrated that Rab proteins work in a highly synchronized fashion. Upon mitochondrial damage and subsequent accumulation of PINK1 on the OMM followed by Parkin recruitment, RabGEF1, which is a GEF and upstream factor of endosomal Rab proteins, is recruited to mitochondria. RabGEF1 recruitment to mitochondria is ubiquitin dependent, with the aid of Parkin. RabGEF1 then directs the localization of Rab5 to mitochondria. Rab5 directs the recruitment of Rab7 via the MON1/CCZ1 complex. To complete Parkin-dependent mitophagy, Atg9A vesicles and LC3-labeled autophagic membranes are assembled on mitochondria in a Rab7-dependent manner, which is inhibited upon Rab7 knockdown. Rab7 GAPs TBC1D15 and TBC1D17 help dissociate Rab7 from the mitochondrial membranes. Another study reported C5orf51 as a novel component of the MON1/CCZ1 complex and was shown to mediate Rab7A translocation to depolarized mitochondria. In the absence of C5orf51, recruitment of Rab7A and Atg9A to mitochondria was inhibited and compromised mitophagy [96]. Interestingly, C5orf51 was identified as a susceptibility gene in AD [97]. Future studies exploring Rab7 and C5orf51 may help to understand the molecular mechanisms in disease pathologies. Another study has reported that Rab7 is involved in mitophagosome formation in Parkin-mediated mitophagy. Rab7 mitochondrial localization and activity states were shown to be regulated by another GAP, TBC1D5, and the retromer complex. Indeed, mitophagy was impaired by the knockout of the retromer complex subunits VPS29 or VPS35 and TBC1D5, and the localization of Rab7 shifted to the lysosomal membranes [98]. Rab7 phosphorylation by TANK-binding kinase 1 (TBK1) is also critical for mitophagosome formation, as cells with non-phosphorylatable Rab7 (mutant Rab7) did not undergo mitophagy and failed to recruit Atg9-positive vesicles to damaged mitochondria. TBK1-mediated phosphorylation of Rab7 at Serine 72 is similar to LRRK2-mediated phosphorylation of its Rab substrates [99]. Since TBK1 is a known component of the cyclic GMP-AMP synthase (cGAS)-Stimulator of IFN genes (STING) pathway in the innate immune response and PINK1/Parkin-mediated mitophagy prevents STING-mediated inflammation induced by mitochondrial impairment [100], exploring Rab7 at the nexus of neuroinflammation and mitophagy in NDs will be tempting. Direct evidence from *PARK2*-mutated fibroblasts has shown a link between Rab proteins and mitochondrial dynamics in the pathogenesis of PD. By using shotgun proteomics and systems biology analyses, Zilocchi and colleagues showed Rab proteins’ involvement in mitochondrial proteome alterations of fibroblasts from *PARK2*-mutated PD patients compared to control subjects. In particular, the study reported Rab7 recruitment to dysfunctional mitochondria without PINK1 accumulation [40]. Another study on GBA1-mutant PD neurons described prolonged mitochondria-lysosome contact sites due to low levels of TBC1D15 and consequent defective Rab7-GTP hydrolysis [101]. Moreover, mutations in Rab7A are associated with CMT2B neuropathy [102], and the disease is also characterized by mitochondrial dysfunction [103]. Future work focusing on Rab7 in mitochondrial dynamics may provide therapeutic targets for these NDs.

Several other Rab proteins have been identified as the downstream players of PINK1. With the aid of SILAC-based phosphoproteomic screening, a recent study reported several Rab substrates of PINK1. Rab8A, Rab8B, and Rab13 were found to be phosphorylated at Serine 111 in a PINK1-regulated manner and independent of Parkin. The latter phosphorylation impairs their activation by GEFs [104]. In fact, PINK1- and LRRK2-mediated phosphorylation of Rab8 are antagonistic to each other. PINK1-mediated phosphorylation of Rab8 at Serine 111 prevents Rab8′s phosphorylation at Ser 72 by LRRK2 [105]. The interplay of PINK1 and LRRK2 in PD and their antagonistic effects in mediating Rab8 activity may provide another interesting nexus in PD pathogenesis. The role of Rab10 in mediating mitophagy was recently reported, where Rab10 accumulated on depolarized mitochondria and recruited OPTN in a PINK1/Parkin-dependent manner. In *LRRK2*-mutated fibroblasts, PINK1 and Parkin activities were not affected, whereas the recruitment of Rab10 and OPTN, mitophagy processes, and mitochondrial function were impaired [106]. Phosphorylated Rab10 (pRab10), was recently reported as a pathological hallmark in AD, where pRab10 was observed to be associated with neurofibrillary tangles in hippocampal neurons. Moreover, pRab10 co-localized with hyperphosphorylated tau (pTau) in the same patients [107]. Rab35, involved in regulating endocytosis, has also been implicated in the classical mitophagy pathway. Along with TBK1, Rab35 promotes NDP52 targeting to damaged mitochondria during mitophagy [108].

Taken together, these findings indicate that the regulation of Rab proteins by their respective GAPs/GEFs and their interaction with effectors and kinases are important steps in the mitophagy process. These studies also provide important links between PD-related genes (*PINK1*, *PARK2*, and *LRRK2*) and Rab proteins, as aberrations in these pathways can possibly be linked to PD.

It is quite evident that multiple quality-control pathways exist to eliminate dysfunctional mitochondria that avoid cell death due to accumulation of these damaged organelles. Many pathways have been uncovered, and certainly more unidentified pathways exist in the cells. Interestingly, Rab proteins are also involved in non-canonical mitophagy.

### 3.2. The Role of Rab Proteins in Non-Canonical Mitophagy

Several mitochondrial quality-control pathways exist to ensure mitochondrial health maintenance. Many studies have reported the clearance of damaged mitochondria by the endosomal system involving Rabs and their specific effector proteins. At times, this is achieved quickly to avoid detrimental accumulation of dysfunctional mitochondria and ultimate cell death. As shown in a recent study [109], depolarized mitochondria after FCCP treatment are sequestered in Rab5-positive endosomes via the endosomal sorting complexes required for transport (ESCRT) machinery. Although Parkin-dependent, this mitochondrial engulfment is different from the classical autophagic engulfment. Indeed, the early endosomes mature into late endosomes, where Rab5 is swapped for Rab7. These mature, late, Rab7-positive endosomes then deliver the damaged mitochondrial cargo to lysosomes for degradation. This study also showed that the early endosomal response to mitochondrial damage precedes the autophagy pathway, whereby the endolysosomal system acts as a first line of defense in clearing damaged mitochondria. Moreover, autophagic activity was increased upon Rab5 abrogation, suggesting that the autophagic pathway compensated for impaired endosomal activity [109]. However, the role of Parkin in mediating the process is controversial. In this study, mitochondrial degradation in both WT and Atg5-/- MEFs was Parkin-dependent [109], which was in contrast to a previous study by Narendra and colleagues reporting that, upon CCCP treatment, Parkin did not mediate damaged mitochondria removal in Atg5-/- MEFs [36].

Furthermore, proteins involved in receptor-mediated mitophagy seem to collaborate with Rab proteins for proper mitochondrial disposal by alternative pathways. Indeed, Hammerling and colleagues [110] showed that BNIP3 promoted damaged mitochondria sequestration into Rab5-positive endosomes independently of autophagy. This study utilized autophagy-deficient Atg5-/- MEFs and showed that, upon FCCP treatment, a significant number of mitochondria accumulated inside the Rab5-positive early endosomes. The mitochondrial content was confirmed by proteomics analysis after isolation of Rab5-positive endosomes [110].

Another model of the mitochondrial quality-control pathway involving Rab5 was reported recently. Hsu and co-workers [111] showed that the endocytic system is the first responder and protector of mitochondria under oxidative stress. This mechanism seems, again, to be distinct to the classical mitophagic pathway. Indeed, this study employed human cancer cells and used laser-induced oxidative damage or H_2_O_2_-induced damage to mitochondria. Interestingly, Rab5 translocated to mitochondria from early endosomes, and this recruitment depended on its effector Alsin, which is implicated in ALS. Once recruited to mitochondria, Rab5 led to the formation of membrane contact sites between the early endosome and mitochondria, inhibited cytochrome *c* release, decreased mitochondrial oxygen consumption, and increased cell viability. An important finding to highlight in this study is that Rab5 recruitment to mitochondria was rapid and preceded autophagy, as this was Parkin-independent, implying a protective role for mitochondria rather than their degradation [111].

A recent study has shown Rab5 to facilitate early endosome-mediated mitophagy [112]. The endosomal adaptor protein APPL1 and Rab5 were shown to localize to damaged mitochondria to promote their elimination and in turn also limit inflammasome activation in bone-marrow-derived macrophages from *APPL1* KO mice. The study also showed that defects in this early endosome-mediated mitophagy led to an accumulation of damaged mitochondria and increased mitochondrial ROS and oxidized mtDNA. These events in turn led to increased inflammasome activity [112]. Collectively, these studies have demonstrated the involvement of Rab5 in different pathways of mitochondrial quality control, where (i) Rab5 was involved in sequestering damaged mitochondria into endosomes leading to their disposal, (ii) Rab5 translocated itself to damaged mitochondria to mediate their removal, and, in other instances, (iii) Rab5 was involved in a rescue pathway to protect mitochondria rather than degrade them.

Mitochondria are also capable of eliminating undesired components by the formation of MDVs which can be PINK1/Parkin-dependent but also occur independently. This alternative disposal pathway has shown the involvement of Rab9, a late endosomal GTPase [113]. In the formation of MDVs, Rab9 is recruited to mitochondria, and Rab7 is then involved in the fusion of MDVs with endosomes. MDVs formation was also stimulated upon heat stress and PINK1 knockdown [113]. Another study reported that mitophagy was severely suppressed upon knockdown of Rab9 and that alternative mitophagy required Rab9, MAPK1, and MAPK14 [114].

The alternative disposal pathways described here likely exist to cope with different types of mitophagy-inducing stressors and/or different doses of the same stressor. Moreover, in different cell types there are various effectors that can associate with specific Rab proteins. In this frame, an unbalanced orchestration of such pathways has severe detrimental effects in neurons, where specific sub-cellular compartments rely on proper mitochondrial activity, such as synaptic terminals. Recently, a clear role for Rab proteins has been unveiled in synaptic damage, which characterizes the very early stages of both sporadic and familial PD [115]. Indeed, this early synaptopathy drives retrograde terminal-to-cell body degeneration, leading to neuronal cell death. Specifically, Rab proteins were found to be involved in the maintenance of synaptic architecture and function, and their dysregulation is implicated in synaptic alterations concerning α-synuclein and LRRK2 pathology [115]. Oxidative phosphorylation is the main process that powers neuronal activities [116]; thus, the lack of functional mitochondria in the right place, namely, at the active synapse, can impair several pathways, such as neurotransmitter release, calcium buffering, and the action potential propagation. It has been demonstrated that synaptic damage may be a leading event in triggering neuronal death [117]. Therefore, the peculiar bioenergetics of neurons should be carefully considered when taking into account the role of Rab proteins in mitochondrial dynamics. As we have already presented, Rab proteins are involved in mitochondrial trafficking and the disposal of dysfunctional organelles. Therefore, they can strongly contribute to energetics failure in neurons. Moreover, autophagy/mitophagy defects can per se cause neurodegeneration by depleting NAD pools [118].

Collectively, this body of evidence highlights the central role of Rab proteins in orchestrating the vesicle trafficking related to the maintenance of a healthy pool of functional mitochondria. Knowledge about the role of Rab proteins in physiological and pathological processes linked to mitochondrial dynamics has been rapidly growing in recent years. However, their functions and interaction partners in common molecular pathways are still to be clarified. Table 1 summarizes currently available evidence that links Rab proteins, mitochondrial dynamics, and neurodegenerative processes.

## 4. Conclusions and Future Perspectives

The recent literature has clearly demonstrated a role for Rab proteins in mitochondrial quality-control pathways, where different stressors and cell types warrant the involvement of specific Rab proteins along with their respective effectors. Understanding the dynamics of Rab proteins in mitochondrial health states is imperative to comprehend whether a dysregulation in Rab dynamics and vesicular trafficking is a causal event in neurodegeneration, particularly in PD. Moreover, the role of Rab proteins in the early stages of the pathogenetic process has now been recognized in several NDs, including AD, where the malfunction of Rab proteins has been linked to protein dyshomeostasis and seems to represent a good candidate for the design of novel therapeutics [119].

This should be coupled with the understanding of the role of impaired mitochondrial disposal in neuronal cells. Long considered an outcome of stress insult accumulation, dysfunctional mitochondria persistence seems to trigger neuron death. The process may start at the synapse, contributing to reduced neuronal functionality, and eventually lead to neurodegeneration. Future studies elucidating the succession of these molecular events will be crucial to better describe the pathogenetic processes of NDs and will allow the identification of potential therapeutic targets.

## Figures and Tables

**Figure 1 ijms-24-06268-f001:**
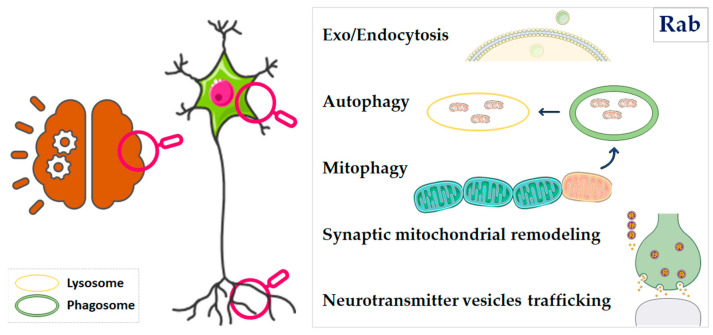
The role of Rab proteins in vesicle trafficking and mitochondrial dynamics. The neurodegeneration process typically involves specific neuronal populations that undergo severe molecular defects ultimately leading to cell death. Rab proteins play a pivotal role in orchestrating several pathways at the level of both neuronal cell body and synaptic terminals. Indeed, they are involved in the formation and trafficking of endosomal vesicles, they participate in the autophagic process by recruiting the membrane machinery, they are directly involved in the mitophagic process and in the maintenance of a healthy pool of mitochondria at the level of the synapse, and eventually they are part of the process of release of neurotransmitter vesicles.

**Figure 2 ijms-24-06268-f002:**
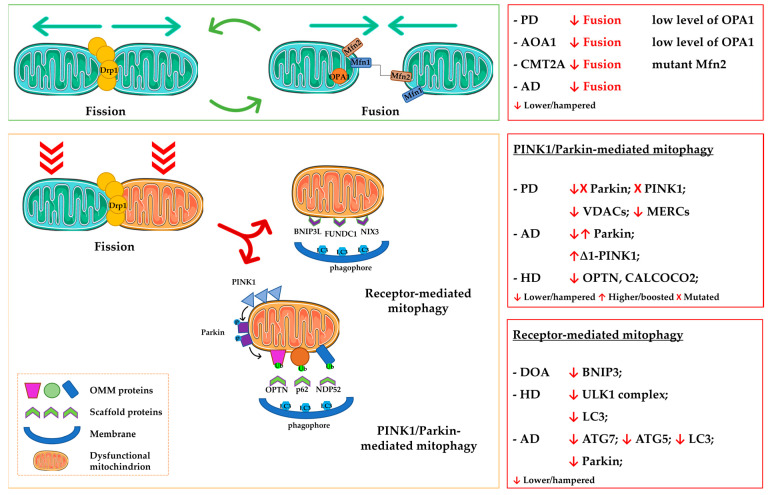
Mitochondrial dynamics defects in neurodegenerative diseases. **Upper** panels: Key molecular players involved in the proper regulation of fusion and fission (**left**) and alterations detected in NDs (**right**). **Bottom** panels: Mechanisms of dysfunctional mitochondria removal, namely, PINK1/Parkin-mediated mitophagy and receptor-mediated mitophagy, with the main molecular factors involved (**left**) and their alterations detected in NDs (**right**).

**Table 1 ijms-24-06268-t001:** Rab proteins involved in mitochondrial dynamics altered in NDs.

Rab Name	Pathways (Interactors)	References
Rab5A	Parkin-dependent micromitophagy via ESCRT machinery (BNIP3)	[109,110]
	PINK1/Parkin-mediated mitophagy (RABGEF1/RABEX-5)	[94,95]
	Early endosome-mediated mitophagy (Rab5 and APPL1)	[112]
	Rab5, along with its effectors GEFs Alsin, implicated in ALS, and Rabenosyn-5, translocate to mitochondria upon stress to patch up damage, preceding mitophagy	[111]
Rab7	PINK1/Parkin-mediated mitophagy (MON1/CCZ1, TBC1D15, and TBC1D17)	[94,95]
	PINK1/Parkin-mediated mitophagy via TBK1 phosphorylation of Rab7a	[99]
	C5orf51 mediates Rab7 localization to mitochondria during mitophagy	[96]
	Rab7 activity and localization to mitochondria regulated by TBC1D5 and the retromer complex	[98]
	Prolonged mitochondria-lysosome contacts in *GBA1*-mutant PD neurons due to defective Rab7-GTP hydrolysis (TBC1D15)	[101]
Rab9	Alternative mitophagy requires Rab9, MAPK1, and MAPK14	[114]
	MDV-mediated damaged cargo removal	[113]
Rab10	Rab10-mediated OPTN accumulation on mitochondria impaired due to *LRRK2* mutations	[106]
Rab35	TBK1-mediated targeting of NDP52 to damaged mitochondria	[108]

## Data Availability

Not applicable.

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
