# Peer review of "The Role of Rab Proteins in Mitophagy: Insights into Neurodegenerative Diseases"

_ijms, 2023, doi:10.3390/ijms24076268_

Round 1
Reviewer 1 Report
Shafique, Brughera et al, reported a comprehensive review of Rab proteins in mitophagy and possible connections to neurodegenerative diseases. Overall, the authors have done a thorough and extensive work of summarizing the Mitochondrial dynamics altered in neurodegeneration, providing a detailed and well-organized overview of the relevant literature and the latest research done.
Despite this, I recommend some minor modifications and interconnections to improve and enrich the manuscript.
· I suggest improving the section " Rab proteins at the crossroads of mitochondrial dysfunction and neurodegeneration" by adding some information, if known, about how remodelling of bioenergetics in neurodegenerative diseases could affect mitochondrial
architecture and dynamics and interplay with Rab proteins. For instance, it is widely described that neurons lack the robust compensatory mechanisms seen in cancer cells, which enable a shift to glycolytic metabolism while maintaining an efficient supply of energy (there are several works and reviews on this aspect such as 1) Hall CN, Klein-Flügge MC, Howarth C, et al. (2012) Oxidative phosphorylation, not glycolysis, powers presynaptic and postsynaptic mechanisms underlying brain information processing. J Neurosci 32: 8940-8951. doi: 10.1523/JNEUROSCI.0026-12.2012 2) Kasischke KA, Vishwasrao HD, Fisher PJ, et al. (2004) Neural activity triggers neuronal
oxidative metabolism followed by astrocytic glycolysis. Science 305: 99-103. doi: 10.1126/science.1096485.
How these aspects could be linked to Rab proteins machinery in mitochondrial dynamics?
· Could the authors briefly describe some research reporting how mitophagy could lead to an adaptive response in neurodegeneration? (See for example Autophagy promotes cell survival by maintaining NAD levels T Kataura, L Sedlackova, EG Otten, R Kumari, D Shapira, F Scialo, Developmental Cell 57 (22), 2584-2598. e11).
· Could the authors mention more clearly the link to how phosphorylated Rab10 may play a role in the pathology and accumulation of NFTs and also cite some Leucine-rich repeat kinase 2 mutations reported as a common genetic cause of AD, such as Zhao, Y.; Ho, P.; Yih, Y.; Chen, C.; Lee, W.L.; Tan, E.K. (LRRK2 variant associated with Alzheimer’s disease. Neurobiol. Aging 2011, 32, 1990–1993)?
· The authors may want to consider including a section on possible
therapeutic targets. Even though, as with all central nervous system diseases, the development of effective and bioavailable pharmacological drugs is highly challenging, the authors could briefly mention some potential pharmacological targets or current therapeutics used or under investigation or approved by EMA or FDA, if any. I believe adding this section could make the review more complete. Here there are some examples "Jordan, K.L.; Koss, D.J.; Outeiro, T.F.; Giorgini, F. Therapeutic Targeting of Rab GTPases: Relevance for Alzheimer’s Disease. Biomedicines 2022, 10, 1141. https://doi.org/10.3390/biomedicines10051141."
· In order to make the review more appealing and help readers identify more easily the roles of the different Rab proteins, I recommend adding a visual abstract in the introduction section, to briefly show and summarize the role and intracellular pathway/ activation of Rab GTPases described. This could be a valuable tool for describing the potential mechanisms underlying the observed associations/ mitochondrial outcomes described in the current Figure 1.
· The authors should consider avoiding some repetition and rephrasing some of the sentences to improve clarity. For instance, I noticed some redundancy such as the word “another” repeated several times throughout the paper.
Author Response
Shafique, Brughera et al, reported a comprehensive review of Rab proteins in mitophagy and possible connections to neurodegenerative diseases. Overall, the authors have done a thorough and extensive work of summarizing the Mitochondrial dynamics altered in neurodegeneration, providing a detailed and well-organized overview of the relevant literature and the latest research done.
Despite this, I recommend some minor modifications and interconnections to improve and enrich the manuscript.
- I suggest improving the section " Rab proteins at the crossroads of mitochondrial dysfunction and neurodegeneration" by adding some information, if known, about how remodelling of bioenergetics in neurodegenerative diseases could affect mitochondrial
architecture and dynamics and interplay with Rab proteins. For instance, it is widely described that neurons lack the robust compensatory mechanisms seen in cancer cells, which enable a shift to glycolytic metabolism while maintaining an efficient supply of energy (there are several works and reviews on this aspect such as 1) Hall CN, Klein-Flügge MC, Howarth C, et al. (2012) Oxidative phosphorylation, not glycolysis, powers presynaptic and postsynaptic mechanisms underlying brain information processing. J Neurosci 32: 8940-8951. doi: 10.1523/JNEUROSCI.0026-12.2012 2) Kasischke KA, Vishwasrao HD, Fisher PJ, et al. (2004) Neural activity triggers neuronal
oxidative metabolism followed by astrocytic glycolysis. Science 305: 99-103. doi: 10.1126/science.1096485.
How these aspects could be linked to Rab proteins machinery in mitochondrial dynamics?
- Could the authors briefly describe some research reporting how mitophagy could lead to an adaptive response in neurodegeneration? (See for example Autophagy promotes cell survival by maintaining NAD levels T Kataura, L Sedlackova, EG Otten, R Kumari, D Shapira, F Scialo, Developmental Cell 57 (22), 2584-2598. e11).
Au: We agree with the reviewer that this is a crucial point. Neuronal cells almost completely rely on oxidative phosphorylation for energy supply and defects in this process cannot be efficiently compensated by the switch to alternative energy production pathways. Therefore, even if already mentioned elsewhere in the manuscript, we decided to stress this point at the end of the 3.2 section. We added the following part (and related references): “Oxidative phosphorylation is the main process that powers neuronal activities [116], thus the lack of functional mitochondria in the right place, namely at the active synapse, can impair several pathways, such as neurotransmitter release, calcium buffering, and the action potential propagation. It has been demonstrated that synaptic damage may be a leading event in triggering neuronal death [117]. Therefore, the peculiar bioenergetics of neurons should be carefully considered when taking into account the role of Rab proteins in mitochondrial dynamics. As we have already presented, the Rab proteins are involved in mitochondrial trafficking and in the disposal of dysfunctional organelles and, as a consequence, they can strongly contribute to the energetics failure in neurons. Moreover, autophagy/mitophagy defects can per se cause neurodegeneration by depleting NAD pool [118].”
- Could the authors mention more clearly the link to how phosphorylated Rab10 may play a role in the pathology and accumulation of NFTs and also cite some Leucine-rich repeat kinase 2 mutations reported as a common genetic cause of AD, such as Zhao, Y.; Ho, P.; Yih, Y.; Chen, C.; Lee, W.L.; Tan, E.K. (LRRK2 variant associated with Alzheimer’s disease. Neurobiol. Aging 2011, 32, 1990–1993)?
Au: Since for AD the role of LRRK2/Rab10 is unrelated to mitochondria, we simply added this sentence (and the related ref) in section 3.1: “Phosphorylated Rab10 (pRab10), was recently reported as a pathological hallmark in AD, where pRab10 was observed to be associated to neurofibrillary tangles in hippocampal neurons. Moreover, pRab10 co-localized with hyperphosphorylated tau (pTau) in the same patients [107].”
- The authors may want to consider including a section on possible therapeutic targets. Even though, as with all central nervous system diseases, the development of effective and bioavailable pharmacological drugs is highly challenging, the authors could briefly mention some potential pharmacological targets or current therapeutics used or under investigation or approved by EMA or FDA, if any. I believe adding this section could make the review more complete. Here there are some examples "Jordan, K.L.; Koss, D.J.; Outeiro, T.F.; Giorgini, F. Therapeutic Targeting of Rab GTPases: Relevance for Alzheimer’s Disease. Biomedicines 2022, 10, 1141. https://doi.org/10.3390/biomedicines10051141."
Au: We agree with the Reviewer that the most challenging aspect of neurodegeneration research is to translate novel molecular discoveries into disease-modifying therapies. However, we think that the addition of a section specifically dedicated to possible therapeutic targets is something that goes beyond the scope of the present review, which is strictly related to molecular factors and pathways that link Rab proteins to mitochondrial dynamics. Thus, we decided to mention in the introductory section 3 the case of LRRK2 inhibitors, which reached clinal trial phase III and represent a promising therapeutic approach: “All pathological LRRK2 mutations lead to hyperactivation of the protein followed by abnormal Rab phosphorylation, which is abrogated upon pharmacological inhibition of LRRK2. Indeed, clinical trials are currently ongoing using small molecules as inhibitors of LRRK2 kinase activity, with potential beneficial effects in both familial and sporadic PD cases. As a downstream effect of LRRK2 inhibition, the restoration of physiological activity of Rab proteins is crucial to explain the therapeutic effect of these drugs [87].”
In addition to this, we also stressed the possibility to exploit Rab-related factors as therapeutics in the Conclusion and Future Perspective section: “Moreover, the role of Rab proteins in the early stages of the pathogenetic process has now been recognized in several NDs including AD, where the malfunction of Rab proteins was linked to protein dyshomeostasis and seems to represent a good candidate for the design of novel therapeutics [119].”
- In order to make the review more appealing and help readers identify more easily the roles of the different Rab proteins, I recommend adding a visual abstract in the introduction section, to briefly show and summarize the role and intracellular pathway/ activation of Rab GTPases described. This could be a valuable tool for describing the potential mechanisms underlying the observed associations/ mitochondrial outcomes described in the current Figure 1.
Au: According to Reviewer’s suggestion, we added an explanatory visual abstract in the introduction section (Figure 1).
- The authors should consider avoiding some repetition and rephrasing some of the sentences to improve clarity. For instance, I noticed some redundancy such as the word “another” repeated several times throughout the paper.
Au: We thank the Reviewer for the kind suggestion. We have removed redundant phrases and words from the manuscript and improved the overall structure for better readability.
Reviewer 2 Report
The manuscript describes the implications of mitochondrial dysfunction and vesicular trafficking in neurodegenerative diseases. I do consider this is a very informative and well written review by the following points:
-Very informative and descriptive introduction. Also well written
-Mitochondrial dynamics are very well described.
-The article is written, and it describes the mechanism very well
Therefore, I consider this manuscript suitable for publication in this journal.
Author Response
The manuscript describes the implications of mitochondrial dysfunction and vesicular trafficking in neurodegenerative diseases. I do consider this is a very informative and well written review by the following points:
-Very informative and descriptive introduction. Also well written
-Mitochondrial dynamics are very well described.
-The article is written, and it describes the mechanism very well
Therefore, I consider this manuscript suitable for publication in this journal.
Au: We thank the Reviewer for the positive comments, and we appreciated that the message was considered clear, well written and already suitable for publication without any further amendments.